# Non-Syndromic Familial Mesiodens: Presentation of Three Cases

**DOI:** 10.3390/diagnostics12081869

**Published:** 2022-08-02

**Authors:** Josefa Alarcón, Jacob Guzmán, Telma S. Masuko, Pablo Navarro Cáceres, Ramón Fuentes

**Affiliations:** 1Research Center in Sciences (CICO-UFRO), Dental School-Facultad de Odontología, Universidad de La Frontera, Temuco 4780000, Chile; josefa.alarcon@ufrontera.cl (J.A.); pablo.navarro@ufrontera.cl (P.N.C.); 2Program of Master in Dental Science, Dental School-Facultad de Odontología, Universidad de La Frontera, Temuco 4780000, Chile; j.guzman04@ufromail.cl; 3Department of Biomorphology, Institute of Health Sciences, Bahia Federal University (ICS-UFBA), Salvador 402331-300, Brazil; tsmasuko@uol.com.br; 4Universidad Autónoma de Chile, Temuco 4780000, Chile; 5Departament of Integral Adult Dentistry, Dental School-Facultad de Odontología, Universidad de La Frontera, Temuco 4780000, Chile

**Keywords:** supernumerary teeth, diagnosis, genetics, complications

## Abstract

Mesiodens are the most common supernumerary teeth and are detected incidentally during routine radiographic examination, so late diagnosis complications are very common. The dentist must make a timely diagnosis and thus avoid clinical complications. Despite advances in knowledge of dental morphogenesis and differentiation, the etiology of mesiodens remains unclear. Therefore, several theories have been postulated to explain how and why they develop. It was described in the literature that heredity could play an important role in the appearance of supernumerary teeth, with a higher rate of appearance in relatives of those affected. This article reports three cases, a mother and two children, who present mesiodens, which shows that supernumerary teeth may involve a genetic factor. In addition, a literature review was carried out to assess the importance of the genetic factor as a possible cause of mesiodens. The relevance and implications of timely diagnosis in clinical practice to avoid manifestations of clinical complications are discussed. Therefore, the identification of the genetic risk factors responsible for the formation of supernumerary teeth is essential for developing a screening tool to determine an individual’s genetic risk.

## 1. Introduction

Supernumerary teeth are a developmental anomaly that is defined as “any tooth or odontogenic structure that forms from tooth buds more than the usual number for any given region of the dental arch” [1,2]. They are also known as hyperdontia and can occur in primary or permanent dentition, be solitary or multiple, be unilateral or bilateral, and affect one or both jaws [3]. Based on their position within the jaws, supernumerary teeth are called mesiodens if they are present in the incisal region, paramolars if they are adjacent to a molar tooth, and distomolars if they are distal to the last molar [4]. Mesiodens are the most common supernumerary teeth [5], occurring in 0.15% to 1.9% of the population [2,6], and are approximately twice as frequent in men as in women [3,7]. Mesiodens can be classified according to their appearance in the permanent dentition (rudimentary mesiodens) or primary dentition (supplementary mesiodens) [8,9] and according to their morphology (conical, tuberculate, or molariform) [8].

In general, an asymptomatic, impacted mesiodens is detected incidentally during routine radiographic examination, because only 25% of maxillary anterior supernumerary teeth erupt [3,10]. Therefore, there must be a high index of suspicion of a mesiodens when its complications manifest, such as asymmetry in the pattern of the eruption of the upper incisors, retained, displaced, or rotated incisors, and the presence of median diastema [1,3,8,10]. In addition, there may be abnormalities in the roots, such as root resorption of the adjacent teeth, as well as the formation of dentigerous cysts [10].

Panoramic, maxillary occlusal, and periapical radiographs are indicated to aid in the diagnosis of mesiodens [3,8]. A panoramic radiograph serves as an aid to detection and provides additional information on associated, missing congenital or supernumerary teeth [8]. However, despite the great utility of panoramic radiography, it only provides two-dimensional information, making cone-beam computed tomography (CBCT) a useful diagnostic tool to identify the precise location and shape of mesiodens without overlaps [10].

Tooth development is an intricate mechanism involving multiple factors, both genetic and environmental, and to fully understand abnormalities of tooth development we must understand the mechanisms and explanations that have been given for these processes over time. There are several theories about the etiology of supernumerary teeth; however, their origin is unknown to date [1,7,9,10,11,12]. (I) The first theory is atavism (phylogenetic theory), which explains supernumerary teeth as an expression of a trait of our simian ancestors, who had more teeth. (II) The second theory is the anomalous division of the dental germ, according to which the follicle is divided into two equal or different parts, which gives rise to two equal teeth or one equal and another dysmorphic. (III) The third is the theory of hyperactivity of the dental lamina; it occurs in the initiation stage of dentition development, and supernumerary teeth are formed as a result of alterations in the hyperactivity of the dental lamina. The last theory is the most widely accepted for the development of supernumerary teeth; however, there is evidence that they can be attributed to environmental factors as well as other factors such as heredity and family tendencies [1,4,9,11].

Many publications focus on the prevalence or treatment of mesiodens supernumerary teeth; however, the etiology of the condition is still unknown. Therefore, the objective of this article is to document three cases of mesiodens, a mother and her children, showing that supernumerary teeth may involve a genetic factor. In addition, a literature review is carried out to assess the importance of the genetic factor as a possible cause of mesiodens. The relevance and implications of timely diagnosis in clinical practice are discussed.

The three cases below differ in the diagnosis of mesiodens according to the time of dental consultation. Authorization to publish the cases was obtained through informed consent and ethical authorization from the Scientific Ethics Committee of the Universidad de La Frontera.

## 2. Cases Presentations

### 2.1. Case Report 1

In 2017, he attended dental care at the Teaching Dental Clinic of the Faculty of Dentistry of the Universidad de La Frontera, Temuco, Chile, as a 9-year-old male patient. The reason for the consultation was dental caries. He did not present any syndrome, systemic disease, or medication; he was a shy but cooperative patient who responded favorably to behavioral management techniques.

In the intraoral clinical examination, the patient presented mixed dentition with multiple cavitated and non-cavitated, active carious lesions. In soft tissue, there was an increase in volume in the hard palate between the upper central incisors (1.1 and 2.1), of hard consistency, painless, with years of evolution according to the mother’s report.

Cone-beam computed tomography was requested for the maxilla (Figure 1). The radiographic study showed the presence of an included supernumerary tooth in a horizontal position in the upper arch, embracing tooth 2.1. The root zone of the mesiodens projected from the midline between teeth 1.1 and 2.1. The crown was located in the upper palatal root third of teeth 2.1 and 2.2. Vestibuloversion of tooth 2.1 was observed (Figure 1).

### 2.2. Case Report 2

In September 2021, the mother of the patient reported in case 1 attended for dental care; she was a 49-year-old female patient with no underlying pathology. Intraoral examination showed partial edentulism, multiple active, cavitated caries, root remains, and discoloration in tooth 2.1, which responded negatively to thermal and percussion tests, with active, cavitated caries on the mesial–palatal–buccal surface. Periapical radiography was requested to plan endodontic treatment in tooth 2.1 (Figure 2) and panoramic radiography to plan rehabilitation treatment (Figure 3). Radiographic examination showed the presence of mesiodens included in a vertical position. Located in the inter-radicular area of teeth 1.1–2.1, the coronal third was in contact with the mesial and palatal surface of the cervical third of tooth 2.1. Mesiodens with the dilacerated apex towards the palate impacted the cortical bone of the nasopalatine canal, with slight compromise of the cortical bone. In addition, a periradicular osteolytic lesion associated with the left central incisor was observed (Figure 2 and Figure 4).

### 2.3. Case Report 3

In October 2021, a 22-year-old woman presented for dental care. Patient had no underlying pathology. There were no signs of any syndrome. On intraoral examination, the patient presented normal, complete permanent dentition, with carious lesions on teeth 1.2 and 1.4. The intraoral soft tissues were healthy, and she practiced good oral hygiene. In the anamnesis, the patient reported a clinical history of the extraction of mesiodens diagnosed by routine radiographic examination 11 years ago. In addition, she reported a history of mesiodens in her mother (case report 2) and her brother (case report 1) Figure 5. No complications were observed after the extraction of mesiodens in permanent dentition (Table 1).

## 3. Discussion

Despite advances in knowledge and differentiation of dental morphogenesis, the etiology of mesiodens remains unclear [1,7,9,10,11,12]. Therefore, several theories have been postulated to explain how and why they develop, the most accepted being the dental lamina hyperactivity theory [1,7,9]. However, based on a literature review conducted between November 2021 and April 2022, we observed that the literature described that heredity plays an important role in the appearance of supernumerary teeth [13,14,15,16,17,18,19,20,21,22,23,24], with a higher rate of appearance in relatives of those affected [3,4,9,11,12] (Table 2).

The databases searched in this literature review were PubMed/MEDLINE, Web of Science, Science Direct, EMBASE, and SCOPUS, using keywords Mesiodens AND Familial Occurrence AND Impacted Teeth AND Etiology, and included primary studies that reported the presence of non-syndromic maxillary mesiodens in subjects without relevant medical history from the same family. The data were extracted from reports of selected cases, and information considered relevant for the analysis are shown in Table 2.

Twelve studies supported the presence of non-syndromic maxillary mesiodens in subjects without relevant medical history from the same family [4,11,13,14,15,16,17,18,19,20,21,22,23,24]. It was associated with an autosomal dominant gene, associated with the X chromosome, and had a higher prevalence in men than in women [25], with a lack of penetrance in some generations [17]. However, cases were described where the presence of mesiodens in continuous generations was reported [21,23].

Kawashima et al. (2006) reported that if either parent exhibited a supernumerary tooth, then the child had a 5989-fold increased risk of developing a supernumerary tooth [11]. Two reports described the presence of mesiodens in father and son [22,23], our report being the first case found in the literature to report the presence of mesiodens in a mother and two of her children. On the other hand, the presence of mesiodens in skipped generations was reported in monozygotic siblings and their grandmother [13].

The literature is abundant on the occurrence of mesiodens in siblings of different ages. The literature search revealed four reports of mesiodens in two siblings [13,18,19,21] and one report of three brothers [16] with the same condition. On the other hand, reports of monozygotic siblings with the presence of mesiodens have been widely described. Townsend et al. [26] reviewed the records of 278 pairs of twins, including panoramic radiographs and dental models. Nine of the twin pairs showed evidence of mesiodens [26]. The literature demonstrates that alterations in dental development can manifest in two ways in the context of this type of twinning. The first manifestation occurs when the features are similar but on opposite sides; in this case, it is called “mirror images” [14,27]. The second manifestation is the expression of a certain identical characteristic in both, called “duplicates”. In the analyzed reports of monozygotic siblings, five cases showed the presence of mesiodens in the same anatomical location [15,17,22,24] and one showed monozygotic twins with mirror alteration [14]. However, cases with a deviation from the patterns described above have also been described, suggesting that the actions of local or environmental factors are also capable of altering the expression patterns of this characteristic [27]. In these case reports, the physical resemblances between some twins were striking, and the many concordant physical features were taken as evidence of monozygosity. These results indicated that genetics may play an important role in the appearance of mesiodens. Our report is consistent with the results exposed by these reports. There was a positive predisposition to mesiodens formation in two double-bonded siblings with the mother exhibiting a mesiodens.

It was reported in the literature that mesiodens are about twice as common in men as in women [3,7]. This is evident in the reported reports; the cases of 29 subjects were reported, of which, 8 were women and 21 were men, indicating that mesiodens is significantly more prevalent in men than in women.

Supernumerary teeth are detected incidentally during radiographic examination since mesiodens rarely erupt within the permanent dentition (about 25%), so late diagnosis complications are very common [24]. It is argued in the literature that the earlier the mesiodens is diagnosed and removed, the better the prognosis [28]. Therefore, the identification of hereditary factors is essential for the development of a screening tool to determine the genetic risk of an individual, to generate an early diagnosis, and, consequently, to prevent further complications [5]. The importance of timely diagnosis is due to the fact that, although supernumerary teeth can sometimes be asympto-matic [23], mesiodens is the type of tooth that causes most of the complications related to included teeth, causing a delay in dental eruption permanent tooth [19,22] and deciduous [17] because they are diagnosed late [23]. In general, mesiodens remain impacted [15], are cone shaped, and are 76–86% single [29]. Cases of single or double mesiodens eruption [13,17,19,20] have also been reported, producing ectopic eruption of the upper incisors, which can cause lesions on the tongue and lodgment of food between the incisors and the mesiodens, leading to difficulty in chewing food [21]. In addition, other complications may occur such as pulp pathologies, tooth rotations, dental caries due to retention of dental bacterial plaque, diastemas [1,3,8,10], and delayed or abnormal root development [13]. Therefore, the knowledge of mesiodens in subjects of the same family is a history of diagnostic suspicion of mesiodens to be considered for early diagnosis and avoiding future complications.

The timing of removal of any unerupted mesiodens is controversial in the literature [28]. Immediate removal versus delayed surgical intervention is advocated after root development of the central incisor and lateral incisor around the age of 8 to 10 years [30]. If postponed after that age, more complex surgical and orthodontic treatment may be necessary. It is recommended to extract the mesiodens in early mixed dentition, which may reduce the need for orthodontic treatments [29]. When mesiodens is detected late and complications are evident, immediate surgical removal is required [30]. For case 1, the supernumerary tooth was in intimate contact with tooth 2.1, which led to vestibuloversion of the left central incisor. Due to the change in position that the mesiodens was generating in the subject of case 1, surgical removal of the mesiodens was decided. However, extraction is not always the treatment of choice for supernumerary teeth. Unerupted mesiodens that do not appear to be affecting the dentition in any way and are found by chance are left in place and kept under observation [5]. For case 2 reported in this report, it was decided not to remove the mesiodens but to follow up at periodic check-ups, because the apical pathology came from a carious lesion, and no complications associated with the mesiodens were reported. Garvey et al. (1999) recommended the monitoring of mesiodens in the absence of associated pathological lesions, when there is no risk of damage to the vitality of the related teeth and when mesiodens occur without symptoms and do not erupt or affect the dentition. These teeth, which are usually found by chance, are best left in place under observation [6]. However, early detection is the best prevention tool for future complications. For case 3, there was no history of the reason for the extraction of the mesiodens, but it was diagnosed late. Therefore, early detection is necessary for timely treatment, and a correct medical and family history could be a fundamental detection tool. Therefore, the value of this case report can be used as a paradigm for the evaluation of the hereditary factors that predispose the patient to the appearance of mesiodens and for the consequent knowledge by the oral surgeon of all the family members in which mesiodens have been detected without any syndromic form. This report presents sufficient evidence to suggest a familial predisposition to produce additional teeth. However, the limitation of this study is that it is mainly based on clinical and radiographic history to describe the familial predisposition to the presence of mesiodens. Future research is encouraged to carry out genetic testing on the patients to corroborate the presence of non-syndromic familial mesiodens.

## 4. Conclusions

This report of three family cases with the presence of mesiodens indicates and corroborates that heredity is involved in the etiology of supernumerary teeth. In general, the detection of mesiodens after the presence of complications can be pretty obvious and late, either due to the patient’s complaint or in the clinical and radiographic examination. Therefore, early and timely detection is a tool that will allow complications to be avoided. This study encourages dentists to identify the presence of mesiodens early through the clinical history of family history of mesiodens, in routine radiographic examination, and during clinical examination. This will allow us to carry out an early approach in the case of patients who do not report signs or clinical symptoms. In addition, it will avoid complications and allow treatment to be planned according to the location and associated anatomical structures of each case.

## Figures and Tables

**Figure 1 diagnostics-12-01869-f001:**
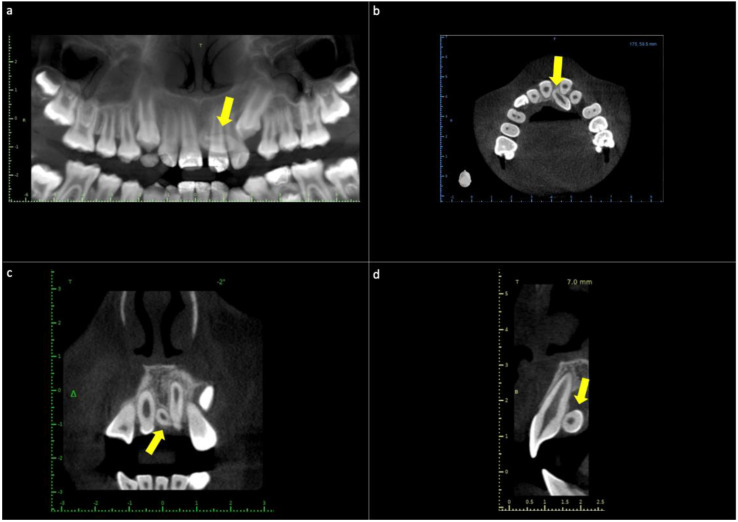
Cone−beam computed tomography (CBCT) of the maxilla. (**a**) The panoramic reconstruction shows the presence of a mesiodens supernumerary tooth between the midlines of maxillary central teeth. (**b**) Axial slice image showing the palatal location of the mesiodens in intimate relation to teeth 2.1 and 2.2. (**c**) Frontal slice image showing mesiodens between central incisors. (**d**) Sagittal slice image showing the palatal location of the supernumerary tooth in the crown-apical middle third of the central incisor. Yellow arrows—location of mesiodens.

**Figure 2 diagnostics-12-01869-f002:**
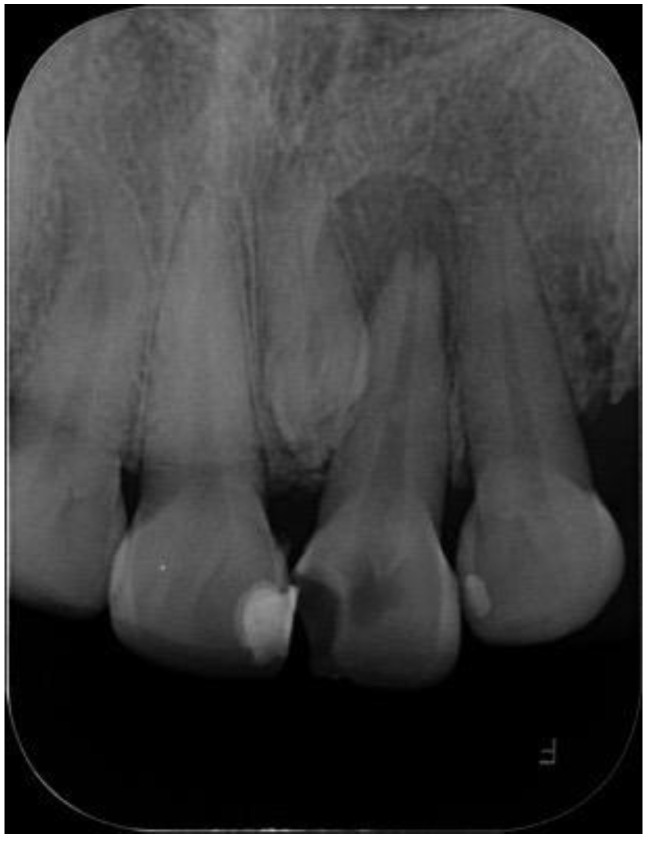
Periapical X-ray showing supernumerary tooth located between tooth 1.1 and 2.1. Mesiodens crown overprojected in mesial root profile of tooth 2.1. Periapical osteolytic radiolucent area with clear non-corticalized limits.

**Figure 3 diagnostics-12-01869-f003:**
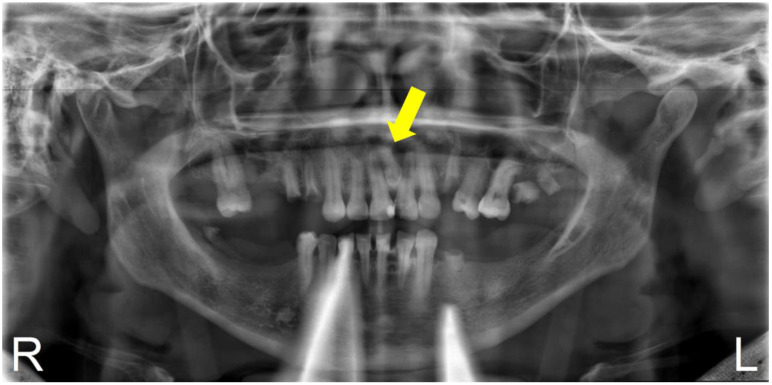
Panoramic X-ray. Bimaxillary partial edentulousness. Presence of multiple root remnants. The presence of supernumerary tooth located between teeth 1.1 and 2.1 is observed. Yellow arrows—location of mesiodens.

**Figure 4 diagnostics-12-01869-f004:**
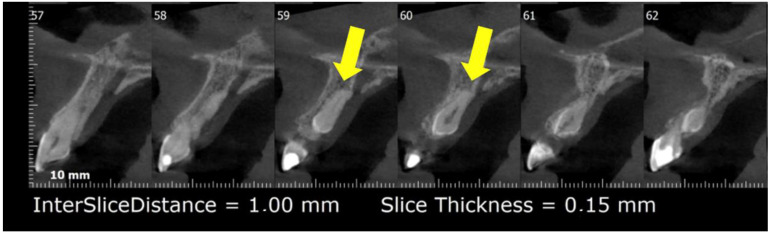
Maxillary cone-beam computed tomography (CBCT). Sagittal slice image. Located in the inter-radicular area of teeth 1.1–2.1. Mesiodens with the dilacerated apex towards the palate, impacted the cortical bone of the nasopalatine canal, with slight compromise of the cortical bone. Yellow arrows—location of mesiodens.

**Figure 5 diagnostics-12-01869-f005:**
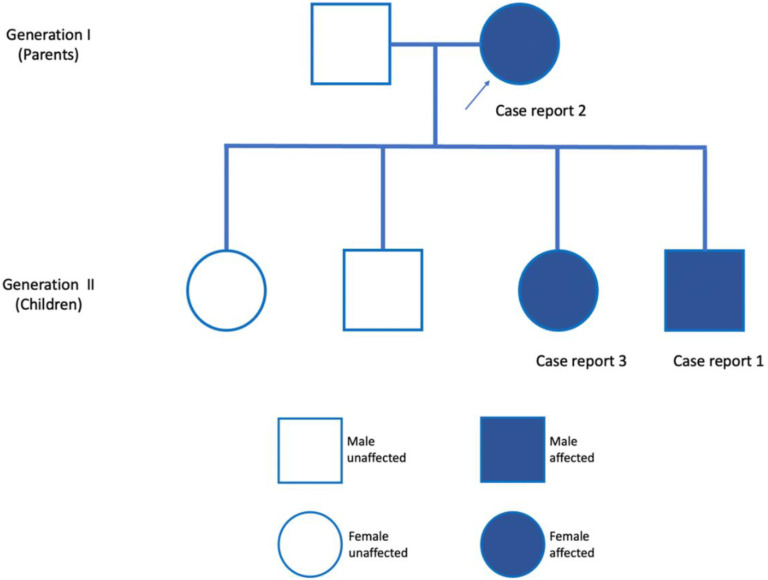
The pedigree shows autosomal dominant transmission of mesiodens in the proband (arrow) and two of her four children.

**Table 1 diagnostics-12-01869-t001:** Case reports.

Case Report	Gender	N° of Supernumeraries	Age at Diagnosis	Diagnostic Form	Complications
Case Report 1	Male	Conoid mesiodens with dilaceration in close contact to tooth 2.1, horizontal position	9 years old	Clinically, due to increased palatal volume	Tooth vestibuloversion 2.1 Palatal volume increase
Case Report 2	Female	Conoid mesiodens, included, vertical position, located between teeth 1.1 and 2.1	49 years old	Radiographic finding	Asymptomatic
Case Report 3	Female	Not reported	12 years old	Radiographic finding	Not reported

**Table 2 diagnostics-12-01869-t002:** Case reports reported in the literature show non-syndromic familial tendency mesiodens.

Study	Family Bond with Supernumerary Teeth	Mesiodens	Age at Diagnosis	Complications
Sedano et al., 1969 [13]	Sister Brother	Sister: Mesiodens inverted Brother: Mesiodens without inversion	Not reported	Sister: The root development was not fully completed Brother: There were no alterations
Beere et al., 1990 [14]	Monozygotic, male	Twin 1: Supplementary primary maxillary lateral incisor on the right side Twin 2: Supplementary primary maxillary incisor on the left side	4 years old	Erupted mesiodens
Choi et al., 1990 [15]	Monozygotic, male	Bilateral mesiodens of opposite orientation Mesiodent roots were fully formed	10 years old	Three conical and impacted mesiodens, one in incisiform eruption
Almeida et al., 1995 [16]	Three siblings	Sibling 1: Presence of an erupted mesiodens Sibling 2: Radiographically, intraosseous mesiodens between the upper incisors were observed Sibling 3: Radiographically revealed an intraosseous development mesiodens	Sibling 1: 11 years old Sibling 2: 9 years old Sibling 3: 21 years old	Sibling 1: Teeth 11 and 21 were poorly positioned Sibling 2: Radiographically detected without complication Sibling 3: Radiographically detected without complication
Seddon et al., 1997 [17]	Monozygotic, male Grandmother	Both twins with bilateral mesiodens Twin 1: The right mesiodens was large with a barrel-shaped crown. The left mesiodens was more incisiform Twin 2: The right mesiodens was conical in form. The left mesiodens was more incisiform Grandmother: Presence of mesiodens	Twins: 7 years old Grandmother: Not reported	Twins: Retained primary central incisors Grandmother: Not reported
Marya et al., 1998 [18]	Two siblings	Sibling 1: Presence of two conical mesiodens eruptions The right mesiodens had a small notehlike projection Sibling 2: Molariform mesiodens eruptions, with five cusps. An intraoral oclusal radiograph revealed the presence of a second, unerupted supernumerary mesiodens	Sibling 1: 12 years old Sibling 2: 8 years old	Sibling 1: Revealed that the maxillary central incisors were positioned far apart because of the presence of two conical mesiodens eruptions. Both permanent maxillary canines were clinically absent Sibling 2: The maxillary left central incisor was malpositioned, the mesiodens was occluding with the mandibular left central incisor and caused a maxillary midline diastema of 4 mm
Gallas et al., 2000 [19]	Two sisters	Both sisters with 2 supernumerary teeth in the position of the central incisors	Sister 1: 13 years old Sister 2: 8 years old	Sister 1: Retention of permanent incisors Sister 2: The permanent central incisors had erupted without the deciduous central incisors having exfoliated
Sharma et al., 2008 [20]	Monozygotic, male	Twin 1: Intraoral examination palatally erupting tuberculate mesiodens in relation to 61. Radiographic examination revealed bilateral presence of two tuberculate mesiodens, one erupted and causing rotation of 21, while the other was seen impacted in relation to 51 Twin 2: Clinical and radiographic examination showed a tuberculate mesiodens between the primary central incisors	7 years old	Twin 1: Complaint of a tooth seen to be erupting for the last 3 months behind the upper front teeth and causing difficulty while speaking Twin 2: Maxillary midline supernumerary tooth which was erupting for the last 2 months
Verma et al., 2010 [21]	Case report 1: sister and brother Case report 2: father and son	Case 1: Intraoral examination revealed two palatally erupting diverging tuberculate mesiodens behind 11 and 21. Intraoral periapical radiograph of 11 and 21 region showed well-defined tuberculate mesiodens with fully formed roots Case 2: Son: A palatally erupted tuberculate mesiodens Father: Fused mesiodens	Case 1 Brother 1: 14 years Brother 2: 11 years Case 2: Son: 11 years old Father: Not reported	Case 1: Mesiodens erupting behind the upper front teeth, revealed recurrent injury to the tongue while making tongue movements. Additionally, there is lodgment of food between upper front teeth and erupting extra teeth leading to difficulty in chewing food Case 2: Son: Tooth erupting behind upper front teeth leading to malalignment of teeth in the upper front teeth region. Displacement of 2.1 Father: Affectation of 1.1
Babacan et al., 2010 [22]	Monozygotic, male	Mesiodens were palatal to the impacted central incisors	10.5 years old	Mesiodens was preventing eruption of the maxillary left central incisor in both twins. Space loss and midline shift of the right central incisor had occurred, and the left lateral incisors had drifted mesially in both children
Sadeghzadeh-Araghi et al., 2019 [23]	Father Son	Father: Mesiodens between roots of central incisors with an unintentional coronectomy, the crown of the tooth had been removed as it was mistaken for a bony growth/extension of the anterior nasal spine Son: Mesiodens was lying behind the anterior nasal spine, buccally positioned and in close proximity to both the floor of the left nasal aperture and maxillary incisor apices	Father: 65 years old Son: 17 years old	Father: Dull ache around the anterior maxilla Son: Radiographic finding, asymptomatic
Witanowska et al., 2011 [24]	Monozygotic twin sisters	Incisiform mesiodens with its root developed; root tip in maxillary midline, mesiodens left tilted, its crown within eruption pathway	8 years 5 months	Permanent upper left incisor: Unerupted, impacted by mesiodens Deciduous upper left incisor: Retained eruption Permanent upper right incisor: Normal eruption These features imply a high degree of co-twin concordance within the region in question

## Data Availability

Not applicable.

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
