# Peer review of "Non-Syndromic Familial Mesiodens: Presentation of Three Cases"

_diagnostics, 2022, doi:10.3390/diagnostics12081869_

Round 1

Reviewer 1 Report

The manuscript is well written and it can be accepted in the present form.

Author Response

Thank you so much for reviewing our work.

Reviewer 2 Report

Dear Authors,

Thank you for submitting your paper to Diagnostics. I hope that my remarks will be useful in order to increase the quality of the paper.

1) I would suggest to reconsider the title. It is interogative and doesn't reflect the content.

2) Line 126 - The picture of the panorex should be replaced with a higher quality one.

3) Lines 134-138 - Please rephrase in an academic manner.

4) The part of literature review is unclear. We cannot talk about a systematic review and you are mixing fruits with vegetables. I would rather recommend to keep the case reports and modify the structure, in such a way, that it will allow to include all the literature findings in the discussion section.

5) The Conclusion section should emphasise the clinical/practical impact of your findings in order to improve further protocols. (Lines 287-299)

Best regards!

Author Response

Thank you so much for reviewing our work. Your suggestions have been highly helpful to improve our manuscript. You may find our reply (R) below each comment (C).

C1.  I would suggest to reconsider the title. It is interogative and doesn't reflect the content.

            R1. Thank you for this comment. El titulo fue reformulado.

C2. Line 126 - The picture of the panorex should be replaced with a higher quality one.

            R2. Thank you for the comment. We replace the picture of the panoex with the imagen highest quality that is available

C3. Lines 134-138 - Please rephrase in an academic manner.

            R3. Thank you for this comment. We rephrase in an academic manner lines 134-138.

C4. The part of literature review is unclear. We cannot talk about a systematic review and you are mixing fruits with vegetables. I would rather recommend to keep the case reports and modify the structure, in such a way, that it will allow to include all the literature findings in the discussion section.

            R4. Thank you for this comment. We modify the structure of the manuscript in such a way that all the findings of the literature are in the discussion section.

C5. The Conclusion section should emphasise the clinical/practical impact of your findings in order to improve further protocols. (Lines 287-299)

            R5. Thank you for this comment. The conclusion was rewritten, to emphasize the clinical/practical.

Reviewer 3 Report

The authors aimed to document 3 cases of mesiodens present in the mother and her children, showing that supernumerary teeth may involve a genetic factor. In addition, a literature review was carried out to assess the importance of the genetic factor as a possible cause of mesiodens. The relevance and implications of timely diagnosis in clinical practice were also discussed.

The study covers some issues that have been overlooked in other similar topics. The structure of the manuscript appears adequate and well divided in the sections. Moreover, the study is easy to follow, but some issues should be improved. Some of the comments that would improve the overall quality of the study are:

a. Authors must pay attention to the technical terms acronyms they used in the text.

b. English language needs to be revised.

c. Limitations of the study needs to be added.

d. Conclusion Section: This paragraph required a general revision to eliminate redundant sentences and to add some "take-home message".

Author Response

Thank you so much for reviewing our work. Your suggestions have been highly helpful to improve our manuscript. You may find our reply (R) below each comment (C).

C1. Authors must pay attention to the technical terms and acronyms they used in the text. 

R1. Thank you for this comment. We carefully proofread the technical terms and acronyms that we used in the text.

C2. The English language needs to be revised.

R2. Thank you for this comment. The English language was revised in the text.

C3. Limitations of the study need to be added.

R3. Thank you for this comment. We included limitations of the study.

C4. Conclusion Section: This paragraph required a general revision to eliminate redundant sentences and to add some "take-home message".

R4. Thank you for this comment. The conclusion was rewritten.

Round 2

Reviewer 2 Report

Dear authors,

Thank you for providing the revised version of the manuscript.

1. Although you have changed the title it doesn't clearly reflect the content of your paper. From my point of you should focus on the case reports part and eliminate "literature review" from the title. You included it in the Discussion section and I believe it's fine. I don't find it useful to mix "case reports" with "reviews" which are basically two different types of papers.

2. Lines 134-141. English revision is required. Maybe you should consider the MDPI support service for Academic English.

3. I still consider that the conclusions section can be improved. Try to focus and emphasise on how the experience that you are sharing can improve the clinical protocols of approaching this case.

Best regards!

Author Response

Thank you so much for reviewing our work. Your suggestions have been highly helpful to improve our manuscript. You may find our reply (R) below each comment (C).

Revisor

Thank you so much for reviewing our work. Your suggestions have been highly helpful to improve our manuscript. You may find our reply (R) below each comment (C).

C1 Although you have changed the title it doesn't clearly reflect the content of your paper. From my point of you should focus on the case reports part and eliminate "literature review" from the title. You included it in the Discussion section and I believe it's fine. I don't find it useful to mix "case reports" with "reviews" which are basically two different types of papers.

R1: Thank you for this comment. The title was reformulated.

C2 Lines 134-141. English revision is required. Maybe you should consider the MDPI support service for Academic English.

R2: Thank you for this comment. The English language was revised in lines 134-141.
C3 I still consider that the conclusions section can be improved. Try to focus and emphasise on how the experience that you are sharing can improve the clinical protocols of approaching this case.

R3: Thank you for this comment. The conclusion was rewritten.

Round 3

Reviewer 2 Report

Dear authors,

Thank you for providing the revised version of your paper.

You responded to my comments so I consider it can be published now.

Best regards!